# BLTR1 Is Decreased in Steroid Resistant Pro-Inflammatory CD28nullCD8+ T Lymphocytes in Patients with COPD—The Spillover Hypothesis Explained?

**DOI:** 10.3390/biology12091261

**Published:** 2023-09-20

**Authors:** Greg Hodge, Hubertus Jersmann, Mark Holmes, Patrick Asare, Eugene Roscioli, Paul N. Reynolds, Sandra Hodge

**Affiliations:** 1Chronic Inflammatory Lung Disease Laboratory, Department of Thoracic Medicine, Royal Adelaide Hospital, Adelaide 5001, Australia; hubertus.jersmann@health.sa.gov.au (H.J.); mark.holmes@health.sa.gov.au (M.H.); patrick.asare@adelaide.edu.au (P.A.); eugene.roscioli@adelaide.edu.au (E.R.); paul.reynolds@health.sa.gov.au (P.N.R.); sandra.hodge@adelaide.edu.au (S.H.); 2Department of Medicine, University of Adelaide, Adelaide 5001, Australia

**Keywords:** COPD, BLTR1, steroid resistant pro-inflammatory lymphocytes

## Abstract

**Simple Summary:**

Certain cells in people’s immune system become inflamed in the lungs of smokers. If they develop emphysema, commonly known as chronic obstructive pulmonary disease (COPD), these inflammatory cells are also present in the blood. The reason why they leave the lungs and enter the blood in COPD patients is currently unknown. When these inflammatory cells enter the blood, they cause further inflammation in other parts of the body, which leads to more medical problems such as cardiovascular disease in patients with COPD. Our studies have shown that due to the increased inflammation in the lungs of COPD patients, these inflammatory cells lose the ability to stay in the lungs due to changes in these cells. We have identified these changes as well as the reasons for these changes. Our studies also show why current medications used to stop inflammation do not work very well on these cells. Now that we know how these cells become resistant to current medications used to prevent inflammation and the reason that they leave the lungs and cause more diseases in patients with COPD, future studies can be planned to target these mechanisms to help improve the health of smokers that develop COPD disease.

**Abstract:**

Introduction: Pro-inflammatory CD8+ T cells are increased in the lungs and also in the peripheral circulation of both smokers and chronic obstructive pulmonary disease (COPD) patients. The reason for this is unclear but has been described as a spillover from cells in the lungs that may cause the systemic inflammation noted in COPD. We have recently shown an increase in steroid-resistant CD28nullCD8+ senescent lymphocytes in the lungs and peripheral blood in COPD. Leukotreine B4 (LB4) receptor 1 (BLTR1) is involved in recruitment of CD8+ T cells to sites of inflammation, and we hypothesized that it may be involved in the migration of these senescent lymphocytes from the lungs in COPD. Methods: Via flow cytometry and Western blot BLTR1, IFNγ, and TNFα expression were measured in peripheral blood, BAL, and large proximal and small distal airway CD28±, CD8± T, and NKT-like cells from COPD patients and healthy control subjects (±prednisolone) following in vitro stimulation. Chemotaxis of leucocyte subsets was determined (±LB4 ± prednisolone). Results: There was an increase in BLTR1-CD28nullCD8+ lymphocytes in the lungs and blood in patients with COPD compared with controls. BLTR1-CD28nullCD8+ T and NKT-like cells produce more IFN/TNF than BLTR+ cells and fail to migrate to LTB4. Treatment with 1 µM prednisolone in vitro resulted in upregulation of BLTR1 expression in pro-inflammatory CD28nullCD8+ cells and migration to LB4. Conclusions: Loss of BLTR1 is associated with an increased inflammatory potential of CD28nullCD8+ T cells and may allow these pro-inflammatory steroid-resistant cells to migrate to peripheral blood. Treatment strategies that upregulate BLTR1 may reduce systemic inflammation and associated co-morbidity in patients with COPD.

## 1. Introduction

COPD is the third leading cause of death worldwide [1]. Existing treatments are largely symptomatic and the only approved anti-inflammatory medication, corticosteroids, has no proven disease-modifying effect [1]. Inhaled corticosteroids have major benefits for the treatment of airway inflammation in asthma, but the reason for their relative lack of efficacy in COPD is both poorly understood and a major limiting factor in COPD treatment. Thus, better understanding of the mechanisms underlying steroid resistance in COPD, and a way to circumvent this to take better advantage of existing therapies, would have an immediate clinical impact.

COPD is a systemic disease and may represent a “spillover” of inflammatory events occurring in the lungs [2], although the available scientific data challenge the spillover hypothesis [3]. In this regard, we have previously shown an increase in pro-inflammatory/cytotoxic T cells, NKT-like cells, and NK cells in the peripheral blood and airways in COPD patients compared with non-COPD smokers, where some changes were only noted in the lungs compared with healthy controls [4]. We have also shown an increase in CD28null senescent CD8+ T and NKT-like cells in the blood of patients with COPD [5]. These cytotoxic, pro-inflammatory lymphocytes showed increased expression of granzyme b/perforin and IFNγ/TNFα compared to their CD28+ counterparts. Importantly, these senescent lymphocytes were resistant to the anti-inflammatory effects of glucocorticoids, possibly due to increased expression of drug efflux pump p-glycoprotein 1 (Pgp-1) [6], decreased expression of glucocorticoid receptor (GCR) [6], histone deacetylase 2 (HDAC2) [6], and nuclear chaperone heat shock protein 90 (Hsp90) [6]. These findings may explain why glucocorticoids (GCS) fail to suppress inflammation in senescent lymphocytes in COPD.

Leukotreine B4 (LTB4) receptor 1 (BLTR1) is involved in the recruitment of CD8+ T cells to sites of inflammation [7]. LTB4 is increased in exhaled breath condensates in COPD patients [8], and LTB4 receptor system is upregulated in the lungs of smokers, who are susceptible to the development of COPD [9].

We have recently shown an increase in CD28+ and CD28nullCD8+ T and NKT-like cells in the small airways in COPD [10] and an increase in CD28nullCD8+ T and NKT-like cells in the blood of patients with COPD [10]. We hypothesized that (a) CD28nullCD8+ T and NKT-like cells migrate to the lungs of COPD patients, and that the migration is facilitated by BLTR1 expression, and (b) once in the airways, the loss of CD28+ expression with subsequent loss of BLTR1 allows for random migration of these cytotoxic, pro-inflammatory cells from the lung into the peripheral blood resulting in systemic inflammatory disease.

## 2. Materials and Methods

### 2.1. Patient and Control Groups

COPD subjects were specifically recruited for the study and informed written consent obtained. No patients had exacerbation of COPD disease for at least six weeks prior to this study. Subjects with other co-existing lung disease or malignancy or aged over 75 years were excluded. Ethics approval was obtained from the Royal Adelaide Hospital Human Ethics Committee. COPD was diagnosed using the GOLD criteria with clinical correlation [Stage I COPD: forced expiratory volume in 1 s (FEV_1_)/forced vital capacity (FVC) < 70% but FEV_1_ ≥ 80% of predicted; Stage II COPD FEV_1_ 50–79% of predicted, Stage III COPD FEV_1_ 30–49% of predicted; Stage IV < 30% of predicted] [11]. COPD subjects were ex-smokers (>1 year) (n = 10) with an average of 39 pack years. No patients were receiving oral GCS. Healthy age-matched non-smoking volunteers (n = 10) with normal lung function and no history of lung disease were recruited as controls. Five of the control group and 4 of the COPD patients had been enrolled in a previous study [10]. However, none were tested at the same time points, and samples were collected between 6 to 24 months later than the previous study. All subjects underwent spirometry as part of their routine clinical assessment. Blood, bronchoalveolar lavage (BAL), and proximal and distal brushings were collected as previously reported [10]. Demographic details of patient and control groups are shown in Table 1. Venous blood was collected into 10 U/mL preservative heparin (DBL, Sydney, Australia), and all samples were maintained at 4 °C until processing. All subjects were submitted to the same protocol and analysis performed retrospectively.

### 2.2. Leucocyte Counts

Full blood counts, including white cell differential counts, were carried out on blood specimens using a CELL-DYN 4000 (Abbot Diagnostics, Sydney, Australia). Blood films and BAL cytospins were stained via the May–Grunwald–Giemsa method and white cell differential counts checked microscopically via morphological assessment. Small airway distal lung brushings were collected as previously reported [10].

### 2.3. CD28± CD8± T, NKT-like Cell Subsets

Aliquots of blood were added to FACS tubes, and red blood cells were lysed using FACSLyse (BD Biosciences, Sydney, Australia) as described previously [3,10].

BAL and proximal and distal brushings samples were centrifuged at 300× *g* for 5 min. After decanting, cells were re-suspended in 2 mL of RPMI completed with 10%FCS (R10, Sigma, Sydney, Australia). BAL macrophages were removed via adherence to plastic dishes for 60 min in a humidified 5% CO_2_/95% air atmosphere at 37 °C, then non-adherent cells were transferred to sterile 10 mL centrifuge tubes and cells and centrifuged at 300× *g* for 5 min; cells were re-suspended in 2 mL of R10. All cells were permeabilized using FACSPerm (BD) as previously reported [3,10], then washed with wash buffer (0.5% BSA in Isoflow (Beckman Coulter, Sydney, Australia)). Appropriately diluted monoclonal antibodies to CD3 perCP.CY5.5 (BD, Sydney, Australia), CD28 PECY7 (BD), CD56 APC (Beckman Coulter, Sydney, Australia), CD8 APC.CY7 (BD), and CD45 V500 (BD) were added for 15 min in the dark at room temperature. After further washing, cells were analysed within 1 h on a FACSCanto II flow cytometer using FACSDiva software V9.0 (BD). Samples were analysed by gating lymphocytes using CD45 staining versus side scatter (SSC) as reported [3,10]. A minimum of 350,000 low SSC events were acquired in list-mode format for analysis. T cells were identified as CD3+CD56-CD45+; CD8 and CD4 T cell subsets were then identified via CD8+ and CD8− staining NKT-like cells which were identified as CD3+CD56+CD45+ low FSC/SSC events [10].

### 2.4. BLTR1, IFNγ, and TNFα Intracellular Cytokine Production by CD28± CD8± T, NKT-like Cell Subsets Using Flow Cytometry

To determine possible association between pro-inflammatory cytokines and BLTR1 expression, CD28±, CD8+, and CD8− T and NKT-like cells; whole blood; BAL; and proximal and distal brushings were treated as previously described, as leucocyte stimulation was required for both intracellular cytokine and BLTR1 expression by T and NKT-like cells. One-mL aliquots of blood (diluted 1:2 with RPMI 1640 medium), BAL, and proximal and distal brushings were treated as described above and placed in sterile 10 mL conical PVC tubes (Johns Professional Products, Sydney, Australia). Phorbol myristate (25 ng/mL; Sigma) and ionomycin (1 μg/mL; Sigma) were added. Brefeldin A (10 μg/mL) was added as a “Golgi block” (Sigma, Sydney, Australia) and the tubes re-incubated in a humidified 5% CO_2_/95% air atmosphere at 37 °C for 16 h. Preliminary experiments showed that addition of brefeldin A had no effect on expression of BLTR1.

Following stimulation and processing, 5 μL of appropriately diluted IFNγ FITC, BLTR1 PE, CD3 perCP.Cy5.5, CD28 PE.CY7, CD56 APC, CD8 APC.CY7, TNFα V450, and CD45 V500 (all BD Biosciences) were added for 15 min in the dark at room temperature. Cells were washed and events acquired and analysed as described above.

### 2.5. BLTR1 Expression in CD28+ and CD28null T Cells via Western Blot

Peripheral blood mononuclear cells (PBMCs) were isolated from the blood of cohorts of control and COPD patients via standard density gradient centrifugation and cells re-suspended at 1 × 10^7^ mL in RPMI 1640 medium. Following stimulation as described above, 5 μL of appropriately diluted CD3 perCP.CY5.5 (BD), CD28 PE.CY7 (BD), CD56 APC (Beckman Coulter), CD8 APC.CY7 (BD), and CD45 V500 (BD) monoclonal antibodies were added for 15 min in the dark at room temperature. Cells were washed and re-suspended in 1 mL RPMI, and CD28+ and CD28null, CD8+, and CD8− T cells were immediately sorted using a FACSAria flow cytometer (BD).

Equal numbers of sorted CD28+ and CD28 null T cells were lysed using M-Per mammalian cell protein lysis reagent with Halt^®^ protease inhibitor cocktail (both Thermo Scientific, Victoria, Australia). Protein samples were quantified using the DC protein assay (Bio-Rad, Victoria, Australia), and 10 μg electrophoresed using Novex^®^ 4–12% gradient Bis-Tris denaturing gels (Life Technologies, Victoria, Australia) and electroblotted to Trans-Blot^®^ Turbo nitrocellulose membrane (Bio-Rad). Membranes were blocked in 5% BSA (Sigma-Aldrich, St Lious, MO, USA), washed, then incubated overnight at 4 °C with anti-human BLTR1 (1:2000) (Sapphire Biosciences, Sydney, Australia. Cat. No. 100019), followed by a 1 h incubation at room temperature with horseradish peroxidase-labelled anti-mouse secondary antibody (R&D Systems, Minneapolis, MN, USA). Chemiluminescent imaging was performed using the LAS-3000 platform, and histogram analysis performed using the Multigauge software package, V3.0 (both FugiFilm, Tokyo, Japan). Mouse-anti Human β-actin antibody (Sigma-Aldrich) was used to correct for loading error for histogram analyses.

### 2.6. Effect of Prednisolone on BLTR1, IFNγ and TNFα Expression by CD28± CD8+ T and NKT-like Cells

To determine the effects of prednisolone on BLTR1, IFNγ and TNFα expression in CD28± CD8 T and NKT-like cell subsets, 1 mL aliquots of PBMCs (diluted 1:2 with RPMI 1640 medium) and distal brushings were placed in sterile 10 mL conical PVC tubes with 1 µM prednisolone for 24 h in a humidified 5% CO_2_/95% air atmosphere at 37 °C. There were insufficient T cells in the BAL and large airway samples to perform these experiments. Blood and small airway cultures were then stimulated as described above for 16 h and assessed for BLTR1, IFNγ, and TNFα expression by CD28± CD8 ± T and NKT-like subsets as described above.

### 2.7. Migration of CD28± CD8+ T and NKT-like Cells

PMBCs were isolated from blood specimens from COPD patients and controls as described above. These cells (50 µL of 1 × 10^7^/mL) were added to the top wells of ChemoTx (96-well Neuroprobe Sydney, Australia) micro-chemotaxis devices (5 µM pore size). Medium ±10 nM LTB4± was added to both top and lower wells. To determine the effects of prednisolone on chemotaxis in CD28± CD8+ T and NKT-like cells, PBMCs were incubated with 1 µM prednisolone for 24 h in a humidified 5% CO_2_/95% air atmosphere at 37 °C before the chemotaxis experiments were performed. There were insufficient lymphocytes obtained from the BAL and proximal/distal brushings to perform these experiments. Spontaneous migration was determined with cells in medium on the top of the filter and medium alone on the bottom. The magnitude of the T cell response was calculated as the percentage of cells that had migrated through the chemotactic filter as a percentage of cells added to the top well.

Preliminary experiments indicated that random migration of CD28null/CD8+/CD3+ cells was reduced compared with their CD28+ counterparts. As cellular levels of HDAC6 have been shown by Cabrero et al. (2006) [12] to be critical for lymphocyte chemotaxis, we measured levels of HDAC6 in sorted CD28null and CD28+CD8+/CD3+ cells via Western blot analysis using the same methodology as for BLTR1 expression above using anti-human HDAC6 (1:5000) (Sapphire Biosciences, Sydney, Australia. Cat. No. 100019), followed by a 1 h incubation at room temperature with horseradish peroxidase-labelled anti-mouse secondary antibody (R&D Systems, Minneapolis, MN, USA).

### 2.8. Statistical Analysis

Statistical analyses were performed using the Mann–Whitney U-test to compare 2 independent samples from control and COPD groups throughout this study. For T cell subsets CD28null/CD8+/CD3+/CD56−/CD45+/TNFα+/IFNγ+), a sample size of n = 10 allowed a power of 98–99.5% for analysis. Variance was estimated from our previous studies [3,10]. Correlations were performed using Spearman Rho correlation tests and differences between groups of *p* < 0.05 considered significant. SPSS software V 9.0 was applied and differences between groups of *p* < 0.05 considered significant.

## 3. Results

### 3.1. Percentages of Blood Lymphocyte Subsets

We noted a significant decrease in peripheral blood T cells in patients with COPD compared with control group (COPD 63.6% of total lymphocytes (31–86) and control 84.0% (71–89), (median (range)). A significant increase in peripheral blood NKT-like cells in patients with COPD compared with control group was identified (COPD 21.3% of total lymphocytes (6–39) and control 6.0% of total lymphocytes (1–26)). There was no change in the percentage of peripheral blood NK cells between groups.

An increase in CD8+ and a decrease in CD4+ T cells in COPD compared with control group (CD8: COPD 75.0% of CD3+ T cells (52–78) and control 48.4% of CD3+ T cells (27–66)) was noted but no changes in any other lymphocyte subsets between groups (*p* > 0.05 for all). These results are consistent with our previous reports [3,10].

### 3.2. Percentages of BAL Lymphocyte Subsets

A decrease in the percentage of BAL T cells and CD4+ T cells in COPD patients compared with control subjects was identified. (CD3+ T cells as a % of total leukocytes: COPD 71% (53–90), control 78% (51–93); CD4+: COPD 34% (31–34), control 66% (12–86)). There was an increase in CD8+ T cells, NKT-like cells, and NK cells in COPD patients compared with control subjects (CD8+ T COPD 63% (58–71) control 33% (13–68); NKT-like: COPD 12% (3–14), control 3% (1–8); NK: COPD 15% (3–21), control 1% (1–4)) (median (range)). These results are also consistent with our previous reports [3,10].

### 3.3. Percentages of Large Airway Brushing Intraepithelial Lymphocyte Subsets

There was no change in the percentage of large (proximal) airway (LA) brushing CD8+ or CD4+ intraepithelial lymphocytes between COPD compared with controls (CD8+ T cells: COPD 74% (56–88), control 67% (31–88) (median (range)); (CD4+ T cells: COPD 22% (15–42) control 31% (15–65) (median (range)). There were no other changes in the percentages of intraepithelial lymphocyte subsets in the LA between groups (*p* > 0.05 for all). These results are also consistent with our previous report [10].

### 3.4. Increased CD28nullCD8+ T and NKT-like Cells in COPD

We noted a significant increase in the percentage of peripheral blood CD28nullCD8+ T cells and CD28nullCD8+ NKT-like cells in COPD patients compared with the healthy control group (Table 2) but no change in CD28+CD8+ T and NKT-like cells. There was no change in the percentage of peripheral blood CD28nullCD8− T cells or NKT cells between the COPD and control groups (Table 2). Similarly, these was a significant increase in the percentage of BAL, large and small airway CD28nullCD8+ T cells, and CD28nullCD8+ NKT-like cells in COPD patients compared with the healthy control group (Table 2) but no change in CD28+CD8+ T and NKT-like cells or CD28nullCD8− (CD4+) T cells or NKT cells between the COPD and control groups (Table 2).

### 3.5. Increased CD28± CD8± T and NKT-like Lymphocyte Subsets Producing IFNγ and TNFα Pro-Inflammatory Cytokines in COPD Patients

There was an increase in the percentage of CD28nullCD8+ T and NKT-like blood lymphocyte subsets producing IFNγ and TNFα pro-inflammatory cytokines in COPD patients compared with the control group (*p* < 0.05 for all) (Table 3).

Similarly, there was an increase in the percentage of CD28nullCD8+ T and NKT-like lymphocyte subsets from BAL, large airway, and small airway producing IFNγ and TNFα in COPD patients compared with the control group (*p* < 0.05 for all) (Table 3). There was a significant increase in the percentage of CD28nullCD8+ T and NKT-like cells producing IFNγ/TNFα in the small airway compared with the large airway, BAL, and blood in patients with COPD (Table 3) (*p* < 0.05 for all).

### 3.6. BLTR1 Expression by CD28± CD8± T and NKT-like Cells

The percentage of CD28nullCD8+ T and NKT-like cells expressing BLTR1 was significantly decreased in blood, BAL, small airway, and large airway from COPD patients and control subjects compared with CD28+CD8+ T and NKT-like cells (Table 4).

### 3.7. Correlation between Small Airway CD28nullCD8+BLTR1-T Cells Producing IFNγ/TNFα and Patient FEV_1_

There was a negative correlation between the percentage of blood and small airway CD28nullCD8+BLTR1-intraepithelial T cells producing IFNγ and TNFα in patients with COPD. Data for blood and small airway IFNγ and TNFα and BLTR1 are shown in Table 3 and Table 4, respectively.

A negative correlation was identified between FEV_1_ and the percentage of small airway and blood CD28nullCD8+ T cells in patients with COPD. Data for FEV_1_ and small airway CD28nullCD8+ T cells are shown in Figure 1.

### 3.8. Effect of Prednisolone on BLTR1, IFNγ, and TNFα Expression by CD28nullCD8+ T and NKT-like Cells

In the presence of 1 µM prednisolone, there was a significant increase in BLTR1 expression by CD28+ and CD28nullCD8+ T and NKT-like cells (T: CD28+CD8+BLTR1+ = 45 ± 15%, CD28nullCD8+BLTR1+ = 115 ± 22%; NKT: CD28+CD8+BLTR1+ = 48 ± 17%, CD28nullCD8+BLTR1+ = 122 ± 25% (median ± SEM n = 4 experiments). The increase in BLTR1 expression was significantly greater for CD28null subsets than for CD28+ T cell subsets (*p* < 0.05 for all). There was a significant inhibition of IFNγ and TNFα expression by all subsets. Inhibition of both IFNγ and TNFα was greater for CD28+ T cell subsets than for CD28null cells (*p* < 0.05 for all). Representative plots showing the effect of 1 µM prednisolone on IFNγ and BLTR1 expression by CD28nullCD8+ T cells are shown in Figure 2.

### 3.9. Chemotaxis of CD28± CD8+ T and NKT-like Cells

There was a significant decrease in the random migration of CD28nullCD8+ T and NKT-like cells in cells suspended in RPMI from the top of the filter with RPMI in the bottom chamber. Data for CD28null and CD28+ T cells are shown in Figure 3. Migration through the filter in response to 10 nM LTB4 in the bottom chamber was significantly increased in CD28+CD8+ T and NKT cells, with no change in CD28nullCD8+ T cell migration (data for CD8 T cells shown in Figure 3). There was no change in the percentage of CD28null and CD28+ T and NKT-like cells migrating through the filter with cells suspended in RPMI+10 nM LTB4 on the top of the filter with RPMI in the bottom chamber (data for CD8 T cells shown in Figure 3). A significant increase in both CD28null and CD28+ T and NKT-like cells migrating through the filter (containing 1 µM prednisolone + RPMI) to the bottom chamber (containing 10 nM LTB4 in RPMI) was shown (data for CD8 T cells shown in Figure 3). There was a significant increase in percentage of CD28+ T and NKT-like cells in the bottom chamber compared with CD28null cells. A significant increase in CD28nullCD8+ T and NKT-like cells migrating through the filter in the presence of 1 µM prednisolone compared with RPMI alone was noted (data for CD8 T cells shown in Figure 3).

### 3.10. BLTR1 Expression of CD28+ and CD28null T Cells by Western Blot

Equal numbers of FACS-sorted CD28+ and CD28null T cells were stained for BLTR1 expression via Western Blot. There was a decrease in the 55 kDa band corresponding to the BLTR1 in CD28nullT cells (white column) compared with CD28+ T cells (grey column) (Figure 4A and Appendix A). BLTR1 expression relative to β-actin from CD28null (CD28−) and CD28+ T cells (median ± SEM from 2 experiments) is shown in Figure 4B.

### 3.11. HDAC6 Expression of CD28+ and CD28null T Cells by Western Blot

Equal numbers of FACS-sorted CD28+ and CD28null T cells were stained for HDAC6 expression via Western blot. There was a decrease in the 130 kDa band corresponding to the HDAC6 in CD28null T cells (white column) compared with CD28+ T cells (grey column) (Figure 4A and Appendix A). HDAC6 expression relative to β-actin from CD28 null (CD28−) and CD28+ T cells (median ± SEM from 2 experiments) is shown in Figure 4B.

## 4. Discussion

The spillover hypothesis has been suggested as a reason for systemic inflammation in patients with COPD [13]. However, the mechanisms behind this phenomenon have not been adequately explained [14]. We have previously shown an increase in CD8+ T cells in the large airways and peripheral blood of patients with COPD, whereas smokers who had not progressed to COPD only showed these changes in the large airways [3]. There have been reports of increased CD8+ T cells in the airways of patients with COPD [15]; however, these previous studies did not investigate expression of CD28 or BLTR1 by these cells. LB4 and BLTR1 have been shown to mediate a potent non-chemokine pathway for cytotoxic T cell trafficking to the lungs [7]. Hence, we hypothesized that this pathway may be involved in the egress of cytotoxic-pro-inflammatory CD8+ lymphocytes from the lungs in patients with COPD. We have previously shown an increase in both CD28+ and CD28nullCD8+ T and NKT-like cells in the large and small airways in patients with COPD and an increase in CD28nullCD8+ T and NKT-like cells in the peripheral blood of patients with COPD compared with smokers who had not progressed to COPD [3,16]. We now show for the first time a loss of BLTR1 from the CD28nullCD8+ T and NKT-like cells in the lungs and peripheral blood of patients with COPD compared with healthy non-smoking volunteers. Importantly we also show BLTR1-CD8+ T cells lose their ability to migrate to LB4. We speculate that cytotoxic effector CD8+ T cells migrate to the lungs, especially the small airways [10], where they encounter persistent activation via inflammatory soluble molecules such as TNFα [17] released from cells from the innate and adaptive immune system (e.g., CD8+ T and NKT-like cells) resulting in loss of the co-stimulatory molecule CD28. This is associated with loss of BLTR1 from these cytotoxic–inflammatory cells with consequent loss of their ability of these cells to stay in the lungs due to high levels of LB4 [18,19,20]. Based on our current findings, we further speculate that the CD28null.BLTR1-CD8+ lymphocytes egress the lungs due to random migration, entering the peripheral circulation and resulting in the systemic inflammatory effects noted in patients with COPD [2]. Interestingly, the random migration of these CD28nullCD8+ T cells was lower than their CD28+ counterparts, suggesting slower egress from the lungs. Cabrero et al. [12] have recently found that cellular levels of HDAC6 are critical for lymphocyte chemotaxis so we performed Western blots on sorted CD28+ and CD28nullCD8+ T cells and identified reduced levels of HDAC6 in CD28nullCD8+ T cells. This may result in these pro-inflammatory cells remaining in the lung longer causing more inflammation. One could speculate that slower egress into the peripheral circulation could result in delayed systemic inflammation caused by these cells. HDAC6 has been shown to be a regulator of HSP90 activity, which we have shown to be reduced in senescent CD28nullCD8+ T cells. There has been a report that glucocorticoids rapidly inhibit cell migration via a novel, non-transcriptional HDAC6 pathway [21]. Further studies comparing HDAC6 levels in these cells from patients receiving glucocorticoids could prove useful in future studies.

Lymphocyte senescence and glucocorticoid resistance occur in several other inflammatory conditions, such as cardiovascular disease [22], autoimmune disease [23], arthritis [24], IBD [25], aging [26], and aging with associated inflammation in COPD [27]. We speculate that the CD28null lymphocytes may be the precursors to these other inflammatory diseases, several of which are potential comorbidities of COPD.

Interestingly, BLTR1 was upregulated in these cytotoxic–pro-inflammatory cells in the presence of the glucocorticoid, prednisolone, a common anti-inflammatory medication used in the treatment of COPD. This finding is consistent with another study showing increased BLTR1 expression on effector memory CD8+ cells in the presence of glucocorticoids [28]. Unfortunately we did not include any patients treated with prednisolone in this current study, but this would be a worthwhile additional study to identify whether COPD patients receiving this anti-inflammatory drug show fewer CD28null.BLTR1-CD8+ lymphocytes in the peripheral circulation (and possibly more CD28nullBLTR1-CD8+ cells in the small airways). If this was shown, these patients may present with less systemic disease than patients not receiving glucocorticoids. Furthermore, there may be an increase in CD28+BLTR1+CD8+ T and NKT-like cell migration into the small airways due to an increase in BLTR1 in these cells in the presence of prednisolone, which may further exacerbate inflammation in the lungs of patients receiving this drug.

Further studies identifying mRNA levels of BLTR1 and inflammatory mediators in CD28null T cells could prove useful in confirming results from this study as well as epigenetic changes associated with senescence and inflammaging in the development of COPD [27].

The relative lack of corticosteroid efficacy in COPD has been poorly understood and a major limiting factor in COPD treatment [2]. We have reported that CD28nullCD8+ lymphocytes respond poorly to the anti-inflammatory effects of steroids due to several mechanisms. The drug efflux pump Pgp-1 is increased in CD8+ lymphocytes in patients with COPD [6]. Furthermore, the glucocorticoid receptor (GCR) is reduced in CD28nullCD8+ lymphocytes in COPD patients, as are nuclear chaperones Hsp90, sirtuin 1, and histone deacetylase 2 [6]. This may result in an increase in the inflammatory effects of CD28nullCD8+ lymphocytes both in the lungs and systemically in these patients.

## 5. Conclusions

In conclusion, we show for the first time that loss of BLTR1 is associated with an increased pro-inflammatory potential of CD28nullCD8+ T cells that may allow these pro-inflammatory steroid-resistant cells to randomly migrate to the peripheral blood. Treatment strategies such as glucocorticoids that upregulate BLTR1 may reduce migration of these pro-inflammatory CD28nullCD8+ T cells into systemic circulation with consequent inflammation and associated comorbidity in patients with COPD.

## Figures and Tables

**Figure 1 biology-12-01261-f001:**
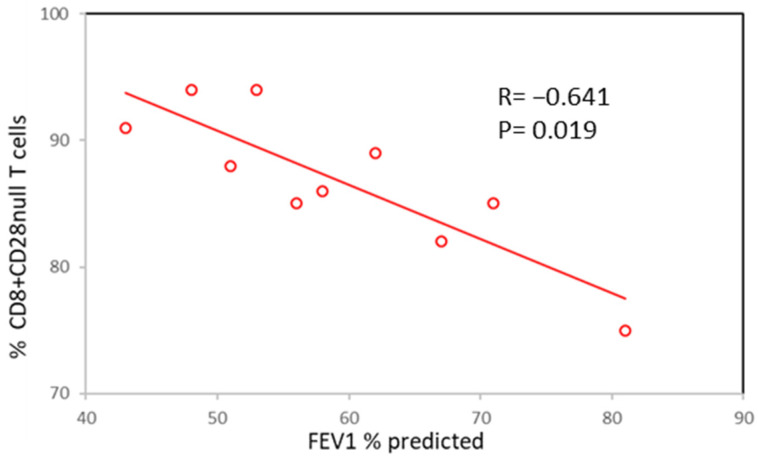
Negative correlation between FEV_1_ and the percentage of small airway CD28nullCD8+ T cells in patients with COPD.

**Figure 2 biology-12-01261-f002:**
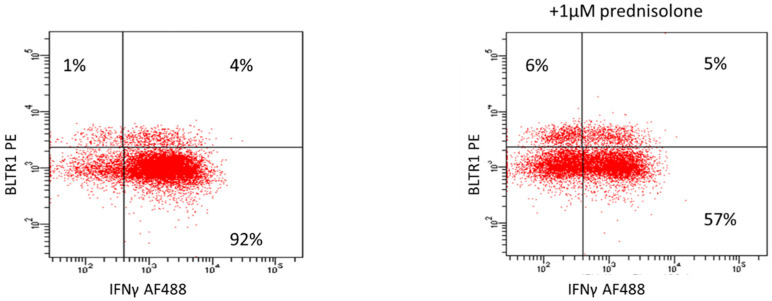
The effects of 1 µM prednisolone compared with medium alone on upregulation of BLTR1 expression by CD28nullCD8+ T and NKT-like cells are shown in Figure 2. In the presence of 1 µM prednisolone, there was a significant upregulation of BLTR1 (*p* < 0.05) and a significant inhibition of IFNγ (*p* < 0.05).

**Figure 3 biology-12-01261-f003:**
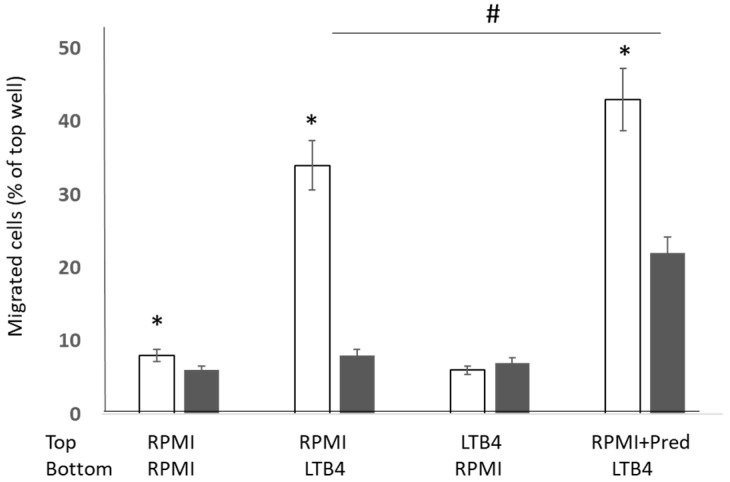
Graph showing the percentage of migrated cells through a 5 µM filter (CD28+CD8+ T cells: clear bars; CD28nullCD8+ T cells: grey bars) in response to ± RPMI ±10 nM LTB4 ± µM prednisolone. There was a significant decrease in the random migration of CD28nullCD8+ T in cells suspended in RPMI from the top of the filter with RPMI on the bottom chamber (*p* < 0.05). There was a significant increase in CD28+CD8+ T cells migrating through the filter in response to 10 nM LTB4 in the bottom chamber (*p* < 0.05) but no change in CD28nullCD8+ T cell migration. There was no change in the percentage of CD28null and CD28+ T and NKT-like cells migrating through the filter with cells suspended in RPMI+10 nM LTB4 on the top of the filter with RPMI in the bottom chamber. There was a significant increase in both CD28null and CD28+ T and NKT-like cells migrating through the filter when cells were incubated with 1 µM prednisolone +RPMI on the top filter with 10 nM LTB4 in RPMI in the bottom chamber. There was a significant increase in percentage of CD28+ T cells migrating through the filter compared with CD28null cells. There was a significant increase in CD28nullCD8+ T cells migrating through the filter in the presence of 1 µM prednisolone compared with RPMI alone (#) (*p* < 0.05). “*”: *p* < 0.05.

**Figure 4 biology-12-01261-f004:**
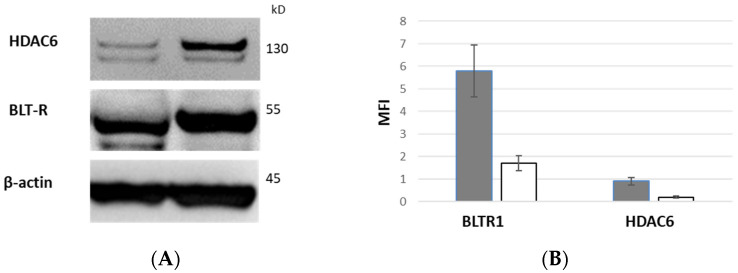
BLTR1 and HDAC6 expression via Western Blot. There was a decrease in the 55 kDa band corresponding to the BLTR1 and 130 kDa band in CD28null T cells compared with CD28+ T cells (**A**). BLTR1 and HDAC6 expression relative to β-actin from CD28 null (CD28−) and CD28+ T cells (median ± SEM from 2 experiments) (**B**).

**Table 1 biology-12-01261-t001:** Demographic details of the study participants.

Subjects	Control	COPD
Number	10	10
Age (years)	55 (43–69) #	56 (36–67)
FEV_1_% pred	103.6 (94.8–102.2)	51 (42–80) *
FEV_1_% FVC	69 (58–71)	50 (34–64) *
Male/female	5/5	5/5

#: median (range). *: significantly decreased compared with control (*p* < 0.05 for all). FEV_1_: forced expiratory volume in 1 s; % pred: percent predicted; FVC: forced vital capacity.

**Table 2 biology-12-01261-t002:** Percentage of CD28nullCD8+ T and NKT-like cells in blood, bronchoalveolar lavage (BAL), small airways (SA), and large airways (LA) from COPD patients and control subjects (as a percentage of CD8+ T cells and CD8+ NKT-like cells). There was a significant increase in CD28nullCD8+ T and NKT-like cells in blood, BAL, SA, and LA from COPD patients compared with control subjects. There was a significant increase in CD28nullCD8+ T and NKT-like cells in SA compared with blood, BAL, and LA from COPD patients.

**COPD**	**T Cells**		**NKT-like Cells**	
	**CD28nullCD8+**	**CD28nullCD4+**	**CD28nullCD8+**	**CD28nullCD4+**
BLOOD	* 54 (36–64) #	7 (3–11)	* 60 (34–71)	5 (3–9)
BAL	* 61 (41–74)	6 (3–12)	* 65 (46–90)	8 (3–10)
LA	* 70 (51–91)	7 (2–12)	* 72 (40–76)	7 (2–12)
SA	* 87 (71–98) @	10 (2–16)	* 89 (74–98)	10 (2–14)
**Control**	**T Cells**		**NKT-like Cells**	
	**CD28nullCD8+**	**CD28nullCD4+**	**CD28nullCD8+**	**CD28nullCD4+**
BLOOD	33 (18–40)	5 (2–10)	34 (21–42)	4 (2–7)
BAL	32 (13–44)	7 (1–14)	35 (19–53)	5 (2–8)
LA	30 (10–41)	8 (3–14)	35 (14–49)	8 (3–14)
SA	37 (20–55)	12 (1–21)	44 (23–58)	11 (3–15)

#: median (range). *: significantly increased compared with control (*p* < 0.05 for all). @: significantly increased compared with blood, BAL, and LA (*p* < 0.05 for all).

**Table 3 biology-12-01261-t003:** Percentage of CD28nullCD8+ T and NKT-like cells producing interferon gamma (IFNγ) and tumour necrosis factor alpha (TNF α) in blood, bronchoalveolar lavage (BAL), small airways (SA), and large airways (LA) from COPD patients and control subjects. There was a significant increase in CD28nullCD8+ T and NKT-like cells producing IFNγ, and TNFα in blood, BAL, SA, and LA from COPD patients compared with control subjects. There was a significant increase in CD28nullCD8+ T and NKT-like cells producing IFNγ and TNFα in SA compared with blood, BAL, and LA from COPD patients.

**COPD**	**T Cells**		**NKT-like Cells**	
	**CD28nullCD8+**		**CD28nullCD8+**	
	**IFNγ**	**TNFα**	**IFNγ**	**TNFα**
BLOOD	* 54 (37–65) #	* 56 (43–67)	* 62 (34–69)	* 65 (39–75)
BAL	* 64 (41–78)	* 61 (41–72)	* 66 (46–88)	* 71 (43–76)
LA	* 71 (51–88)	* 65 (52–86)	* 72 (42–82)	* 79 (50–81)
SA	* 88 (71–98) @	* 93 (76–97) @	* 89 (72–99) @	* 90 (71–96) @
**Control**	**T Cells**		**NKT-like Cells**	
	**CD28nullCD8+**		**CD28nullCD8+**	
	**IFNγ**	**TNFα**	**IFNγ**	**TNFα**
BLOOD	32 (18–40)	34 (23–41)	34 (22–41)	33 (14–46)
BAL	34 (12–44)	32 (13–35)	37 (17–51)	34 (12–44)
LA	30 (14–41)	38 (12–39)	31 (15–44)	38 (13–55)
SA	39 (20–56)	29 (10–42)	41 (26–65)	32 (10–44)

#: median (range). *: significantly increased compared with control (*p* < 0.05 for all). @: significantly increased compared with LA (*p* < 0.05 for all).

**Table 4 biology-12-01261-t004:** Percentage of CD28null and CD28+CD8+ T and NKT-like cells expressing BLTR1 in blood, bronchoalveolar lavage (BAL), small airways (SA), and large airways (LA) from COPD patients and control subjects. There was a significant decrease in CD28nullCD8+ T and NKT-like cells expressing BLTR1 in blood, BAL, SA, and LA from COPD patients and control subjects compared with CD28+CD8+ T and NKT-like cells.

**COPD**	**BLTR+T Cells**		**BLTR+NKT-like Cells**	
	**CD28+CD8+**	**CD28nullCD8+**	**CD28+CD8+**	**CD28nullCD8+**
BLOOD	56 (37–69) #	* 6 (4–9)	60 (38–70)	* 3 (1–4)
BAL	52 (35–66)	* 7 (3–10)	65 (41–77)	* 5 (2–7)
LA	50 (36–61)	* 8 (4–11)	72 (40–78)	* 6 (4–10)
SA	47 (34–63)	* 3 (2–5)	58 (38–72)	* 2 (1–4)
**Control**	**BLTR1+T Cells**		**BLTR1+NKT-like Cells**	
	**CD28+CD8+**	**CD28nullCD8+**	**CD28+CD8+**	**CD28nullCD8+**
BLOOD	62 (38–73)	* 10 (2–10)	62 (37–75)	* 9 (5–13)
BAL	59 (38–71)	* 11 (1–14)	61 (39–73)	* 11 (6–15)
LA	60 (39–72)	* 13 (3–14)	67 (42–78)	* 12 (7–15)
SA	55 (37–68)	* 10 (1–12)	57 (37–70)	* 11 (6–14)

#: median (range). *: significantly decreased compared with CD28+CD8+ (*p* < 0.05 for all).

## Data Availability

All data are present in the manuscript.

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
