# Peer review of "BLTR1 Is Decreased in Steroid Resistant Pro-Inflammatory CD28nullCD8+ T Lymphocytes in Patients with COPD—The Spillover Hypothesis Explained?"

_biology, 2023, doi:10.3390/biology12091261_

Round 1

Reviewer 1 Report

The authors continue their work on CD28± T cells in COPD. Here they show CD28null CD8+ T cells (and NKT cells) are enriched in COPD, particularly in the airway, despite being LTB4 unresponsive by virtue of the relative absence of BLTR1 on these cells as they show. Interestingly, prednisolone treatment markedly improves CD28null CD8+ T cell migratory response to LTB4, despite a relatively modest increase in BLTR1 expression on these cells.

The authors report that IFNgamma is reduced in CD28null CD8+ T cells upon treatment with prednisolone and discuss this in the context of an immunosuppressive effect. However, to make this conclusion, the authors must also assay soluble IFNgamma post-treatment to rule out disbursement by the cells of this cytokine upon treatment, as this would have negative consequences in vivo that might explain why in fact CS treatment is often seemingly neutral at best in COPD.

The abundance of airway lymphocytes reported seems excessive, compared with values at our centre as well as others in COPD, where BAL histology shows 70-90% macrophages. It does seem to hold true that lymphocytes measured by flow cytometry exceed those counted by histology, in our experience and others, for reasons probably beyond discussion in this paper, but the authors report here that a minimum of 50% of their BAL cells are lymphocytes. This does not track with their prior outputs (doi.org/10.1002/cyto.b.20020) where lymphocytes were 55% at maximum. The cells seem to be in excess which might prompt re-examination of the patients chosen as we occasionally note marked lymphocyte elevation in patients with autoimmune comorbidities.

CD28 expression on T cells declines with age, it would be interesting to see whether CD28 positivity correlates with age in either cohort.

The data currently reported in Table 2 seem to match data previously reported, in table 3 of reference 10, by the authors. There are slight differences, though the COPD patient cohorts used look like they may be the same or with substantial overlap. Whether this is an issue depends on journal policy, as it seems that new biological features are being reported using the same cells, leading to novel observations. Relatedly, figure one in the current submission seems to be reproduced from figure 2 in the authors prior study (reference 10). Indeed, the abstract of the current manuscript reports that GCR was measured, but this is not among the data, but was part of the analysis of the prior figure 2.

Minor:

It’s not fully clear what the values in table 2 are a percentage of. I gather that for each compartment we are shown what percentage of T cells are CD28null, with the conclusion for example that 97% of T cells on the small airway surface are CD28null in COPD.

Line 95: In the methods, the authors indicate the cells were permeabilised as an initial step for flow staining, which would be counter indicated for surface staining; do the authors mean instead that the cells were fixed or should this permeabilization step have instead been included in the following section where intracellular cytokines are probed? It is probable that permeabilisation at the outset of staining will lead to an over-estimate of staining and possibly explain the extremes of lymphocyte numbers reported.

Line 160: one symbol too many.

Author Response

biology-2578427

Response to Reviewer 1

The authors continue their work on CD28± T cells in COPD. Here they show CD28null CD8+ T cells (and NKT cells) are enriched in COPD, particularly in the airway, despite being LTB4 unresponsive by virtue of the relative absence of BLTR1 on these cells as they show. Interestingly, prednisolone treatment markedly improves CD28null CD8+ T cell migratory response to LTB4, despite a relatively modest increase in BLTR1 expression on these cells.

The authors report that IFNgamma is reduced in CD28null CD8+ T cells upon treatment with prednisolone and discuss this in the context of an immunosuppressive effect. However, to make this conclusion, the authors must also assay soluble IFNgamma post-treatment to rule out disbursement by the cells of this cytokine upon treatment, as this would have negative consequences in vivo that might explain why in fact CS treatment is often seemingly neutral at best in COPD.

Response: The use of brefeldin A (a golgi block) prevents release of intracellular cytokines from these cells in these experiments (https://www.sciencedirect.com/topics/nursing-and-health-professions/brefeldin-a).

The abundance of airway lymphocytes reported seems excessive, compared with values at our centre as well as others in COPD, where BAL histology shows 70-90% macrophages. It does seem to hold true that lymphocytes measured by flow cytometry exceed those counted by histology, in our experience and others, for reasons probably beyond discussion in this paper, but the authors report here that a minimum of 50% of their BAL cells are lymphocytes. This does not track with their prior outputs (doi.org/10.1002/cyto.b.20020) where lymphocytes were 55% at maximum. The cells seem to be in excess which might prompt re-examination of the patients chosen as we occasionally note marked lymphocyte elevation in patients with autoimmune comorbidities.

Response: We only focussed on lymphocytes in these experiments in blood, BAL and brushings. Line 147..Samples were analyzed by gating lymphocytes using CD45 staining versus side scatter (SSC) as reported [3,10].   Line 135.. BAL macrophages were removed by adherence to plastic dishes…..

CD28 expression on T cells declines with age, it would be interesting to see whether CD28 positivity correlates with age in either cohort.

Response: As there was no difference in the age of patient and control groups we did not analyse for this correlation.

The data currently reported in Table 2 seem to match data previously reported, in table 3 of reference 10, by the authors. There are slight differences, though the COPD patient cohorts used look like they may be the same or with substantial overlap. Whether this is an issue depends on journal policy, as it seems that new biological features are being reported using the same cells, leading to novel observations. Relatedly, figure one in the current submission seems to be reproduced from figure 2 in the authors prior study (reference 10). Indeed, the abstract of the current manuscript reports that GCR was measured, but this is not among the data, but was part of the analysis of the prior figure 2.

Response: Five of the control group and 4 of the COPD patients were enrolled previously in the reference 10 study. However none were tested at the same time points and samples were collected between 6 to 24 months later than the previous study. This may be the reason for similar results between the two studies. Added to text: Line 68: Five of the control group and 4 of the COPD patients were enrolled in a previous study [10]. However none were tested at the same time points and samples were collected between 6 to 24 months later than the previous study. Measurement of GCR has been omitted from the study as the data added nothing new from previously published studies. Line 8: BLTR1, (GCR deleted), IFNγ and TNFα expression were measured……

Minor:

It’s not fully clear what the values in table 2 are a percentage of. I gather that for each compartment we are shown what percentage of T cells are CD28null, with the conclusion for example that 97% of T cells on the small airway surface are CD28null in COPD.

Response: Inserted in line 219: (as a percentage of CD8+ T cells and CD8+ NKT-like cells).

Line 95: In the methods, the authors indicate the cells were permeabilised as an initial step for flow staining, which would be counter indicated for surface staining; do the authors mean instead that the cells were fixed or should this permeabilization step have instead been included in the following section where intracellular cytokines are probed? It is probable that permeabilisation at the outset of staining will lead to an over-estimate of staining and possibly explain the extremes of lymphocyte numbers reported.

Response: We have previously shown that by using the perm step removes any red blood cells that are resistant to FACSLyse following stimulation experiments with PMA/Ionomycin overnight from blood, BAL and brushings making lymphocyte gating strategy easier following exclusion of these contaminants. Percentages of different cell types are unaltered using this step at any stage. (G. Hodge PhD thesis University of South Australia 2000).

Line 160: one symbol too many.

Response: I could not find this symbol.

Reviewer 2 Report

With interest, I read the manuscript biology-2578427.

1.      The data presented here seem to be interesting but the article seems to have very little content. Please, change its type to brief report/short communication.

2.      What about mRNA level expression and epigenetic (see e.g. your references 6 and 25) analyses? Those could be mentioned in limitations and further plans (or lines 424-431 expanded).

3.      Statistics is very unclear. It should be thoroughly discussed with a statistician. Just for example:

a.      Table 1. You mean Mann-Whitney U-test was used here under the name “Wilcoxon”? It is unpaired comparison. Besides, why the name of the test is given in this table only or otherwise in the methods?

b.      Why “Friedman test with Wilcoxon sign rank test”? Are all your comparisons paired? In which comparisons more than two groups are used?

4.      Please, make a graphical abstract to illustrate your findings and their potential meaning.

Round 2

Reviewer 2 Report

A. The article does not have sufficient content for a full-length paper. The Authors should at least try to expand e.g. the Discussion if they have no additional data. So far, the article refers to 26 papers only, so it there is much space to relate the data report in this work to some further papers.

B. There are multiple fonts used throughout the manuscript.

C. Figure 3 vs. Figure 4B: please, unify the style.

D. Graphical abstract would add to this work and expand it as well.  

Author Response

Reviewer 2-Response to R1

Comments and Suggestions for Authors

  1. The article does not have sufficient content for a full-length paper. The Authors should at least try to expand e.g. the Discussion if they have no additional data. So far, the article refers to 26 papers only, so it there is much space to relate the data report in this work to some further papers.

Response: We have added additional data to the manuscript:

Line 173: Preliminary experiments indicated that random migration of CD28null/CD8+/CD3+ cells was reduced compared with their CD28+ counterparts. As cellular levels of HDAC6 have been shown to be critical for lymphocyte chemotaxis (Cabrero et al. (2006) we measured levels of HDAC6 in sorted CD28null and CD28+ CD8+/CD3+ cells by Western Blot analysis using the same methodology as for BLTR1 expression above using anti-human HDAC6 (1:5000) (Sapphire Biosciences, Sydney, Australia. Cat. No. 100019), followed by a 1 h incubation at RT with horse radish peroxidase-labelled anti-mouse secondary antibody (R&D Systems, MN, USA).

Line 380: HDAC6 expression of CD28+ and CD28null T cells by Western Blot

Equal numbers of FACS-sorted CD28+ and CD28null T cells were stained for HDAC6 expression by Western Blot. There was a decrease in the 130 kDa band corresponding to the HDAC6 in CD28null T cells (white column) compared with CD28+ T cells (grey column) (Figure 4A). HDAC6 expression relative to β-actin from CD28 null (CD28-) and CD28+ T cells (median ± SEM from 2 experiments) is shown in Figure 4B.

We combined BLTR1 and HDAC6 western blots:

            Figure 4A                            Figure 4B

Figure 4. BLTR1 and HDAC6 expression by Western Blot. There was a decrease in the 55 kDa band corresponding to the BLTR1 and 130 kDa band in CD28null T cells compared with CD28+ T cells (Figure 4A). BLTR1 and HDAC6 expression relative to β-actin from CD28 null (CD28-) and CD28+ T cells (median ± SEM from 2 experiments) (Figure 4B).

 Line 423: . Cabrero et al.20 have recently found that the cellular levels of HDAC6 are critical for lymphocyte chemotaxis so we performed western blots on sorted CD28+ and CD28null CD8+T cells and identified reduced levels of HDAC6 in CD28nullCD8+T cells. This may result in these pro-inflammatory cells remaining in the lung longer causing more inflammation. One could speculate that slower egress into the peripheral circulation could result in delayed systemic inflammation caused by these cells. HDAC6 has been shown to be a regulator of HSP90 activity which we have shown to be reduced in senescent CD28nullCD8+ T cells. There has been a report that glucocorticoids rapidly inhibit cell migration through a novel, non-transcriptional HDAC6 pathway.21 Further studies comparing HDAC6 levels in these cells from patients receiving glucocorticoids could prove useful in future studies.

Line 550: added the 2 new references

  1. Stephen Kershaw S, Morgan DJ, Boyd J, Spiller DG, Kitchen G, Zindy E, Iqbal M, Rattray M, Sanderson CM, Brass A, Jorgensen C, Hussell T, Matthews LC, Ray DW. Glucocorticoids rapidly inhibit cell migration through a novel, non-transcriptional HDAC6 pathway. Journal of Cell Science (2020) 133, jcs242842. doi:10.1242/jcs.242842
  2. Cabrero JV, Serrador JM, Barreiro O, Mittelbrunn M, Naranjo-Sua ́ rez S, Martı ́n-Co ́ freces N,* Vicente-Manzanares M, Mazitschek R, Bradner JE,§Jesu ́s A ́vila J, Valenzuela-Ferna ́ ndez A, Sa ́ nchez-Madrid F. Lymphocyte chemotaxis Is regulated by histonedeacetylase 6, independently of its deacetylase activity. Molecular Biology of the Cell 2006; 17, 3435–3445.

 Copy of the original western blot data for BLTR1 and HDAC2:

  1. There are multiple fonts used throughout the manuscript.

Response: I think this will be sorted by the production editor?

  1. Figure 3 vs. Figure 4B: please, unify the style.

Response: Will leave this to the editor

  1. Graphical abstract would add to this work and expand it as well.  

Round 3

Reviewer 2 Report

Let it be.